# Unravelling the genetic landscape of cervical insufficiency: Insights into connective tissue dysfunction and hormonal pathways

Ludmila Voložonoka[1,2]*, Līvija Bārdiņa[1,2], Anna Kornete[1,3], Zita Krūmiņa[1], Dmitrijs Rots[1,2], Meilė Minkauskienė[4], Adele Rota[1,3], Zita Strelcoviene[4], Baiba Vilne[1], Inga Kempa[1], Anna Miskova[1,3], Linda Gailīte[1], Dace Rezeberga[1,3,5]

1 Riga Stradins University, Riga, Latvia, 2 Children's University Hospital, Riga, Latvia, 3 Riga Maternity Hospital, Riga, Latvia, 4 Lithuanian University of Health Sciences, Kaunas, Lithuania, 5 Riga East Clinical University Hospital, Riga, Latvia

* ludmilavolozonoka@gmail.com

**Data Availability Statement:** All relevant data are within the manuscript and its Supporting Information files.

## Abstract

### Background

The intricate molecular pathways and genetic factors that underlie the pathophysiology of cervical insufficiency (CI) remain largely unknown and understudied.

### Methods

We sequenced exomes from 114 patients in Latvia and Lithuania, diagnosed with a short cervix, CI, or a history of CI in previous pregnancies. To probe the well-known link between CI and connective tissue dysfunction, we introduced a connective tissue dysfunction assessment questionnaire, incorporating Beighton and Brighton scores. The phenotypic data obtained from the questionnaire was correlated with the number of rare damaging variants identified in genes associated with connective tissue disorders (in silico NGS panel). SKAT, SKAT-O, and burden tests were performed to identify genes associated with CI without a priori hypotheses. Pathway enrichment analysis was conducted using both targeted and genome-wide approaches.

### Results

No patient could be assigned monogenic connective tissue disorder neither genetically, neither clinically upon clinical geneticist evaluation. Expanding our exploration to a genome-wide perspective, pathway enrichment analysis replicated the significance of extracellular matrix-related pathways as important contributors to CI's development. A genome-wide burden analysis unveiled a statistically significant prevalence of rare damaging variants in genes and pathways associated with steroids (p-adj = 5.37E-06). Rare damaging variants, absent in controls (internal database, n = 588), in the progesterone receptor (*PGR*) (six patients) and glucocorticoid receptor (*NR3C1*) (two patients) genes were identified within key functional domains, potentially disrupting the receptors' affinity for DNA or ligands.

**Funding:** The study was funded by the Fundamental and Applied Research Projects grant of The Latvian Council of Science. Project 'Elucidating comprehensive etiology of cervical insufficiency to foster timely diagnosis of preterm delivery and prevent adverse outcomes in obstetrics', Nr. 2020/1-0042 (to DR). The funders had no role in the study design, data collection or analysis, decision to publish, or preparation of the manuscript.

**Competing interests:** NO authors have competing interests.

## Conclusion

Cervical insufficiency in non-syndromic patients is not attributed to a single connective tissue gene variant in a Mendelian fashion but rather to the cumulative effect of multiple inherited gene variants highlighting the significance of the connective tissue pathway in the multifactorial nature of CI. *PGR* or *NR3C1* variants may contribute to the pathophysiology of CI and/or preterm birth through the impaired progesterone action pathways, opening new perspectives for targeted interventions and enhanced clinical management strategies of this condition.

## Introduction

Preterm birth (PTB) is a significant yet poorly understood outcome of pregnancy, representing a major public health concern [1]. The prevalence of PTB is estimated to range from 5% to 18% globally. PTB accounts for approximately 75% of all neonatal deaths [2,3]. Prolonged stays in PTB neonatal-intensive care units for PTB infants, recurrent hospital admissions, and specialized PTB follow-up needs adds of significant financial strain [1,4].

### Preterm birth: One outcome—Multiple aetiologies and distinct molecular pathways

The aetiology of PTB is multifaceted and complex. Various factors, including uterine anomalies, placental abruption, preterm premature rupture of membranes (PPROM), inflammatory conditions (such as intraamniotic inflammation and/or infection), and cervical insufficiency (CI), either individually or in combination, contribute to the occurrence of PTB [5]. Furthermore, genetic predisposition has been recognized as a potential influence on PTB [6,7]. Studies have suggested heritability estimates for PTB ranging from 15% to as high as 30–40% [8].

Findings from the literature collectively support the prevailing hypothesis that PTB is driven through two main pathways: 1) the inflammatory pathway–suggesting that PTB, at least in part, has an inflammatory aetiology. Inflammatory pathway can be induced by infection, or may result from sterile inflammation due to intracellular processes yet to be identified [6]; 2) connective tissue dysfunction pathway–biomechanical properties of the connective tissues can be disrupted by e.g. uterine overdistension, cervical or pelvis tissue laxity leading to preterm cervical dilatation, rendering it impossible to sustain an otherwise healthy pregnancy to full term. Nevertheless, lesser-known pathways and candidate genes within these pathways continue to emerge as significant in relation to prematurity, including, but not limited to: hemopoietic pathway, haemostasis, coagulation, focal adhesion, and cell communication pathways [9,10].

Nonetheless, determining the genetic factors contributing to PTB has been challenging. Despite advancing genetic studies in almost all clinical disciplines, our understanding of the genetic aetiologies and molecular pathways of PTB and PTB-associated phenotypes remains limited. Consequently, our ability to develop interventions to prevent and/or treat PTB by specifically targeting the underlying phenotypes–e.g., CI, PPROM, maternal-foetal immune rejection, etc.–is hindered. Currently, the existing treatment of PTB in clinical practice predominantly rely on observation and symptomatic/prophylactic management (e.g. progesterone, cervical cerclage and pessary) rather than targeting the specific underlying aetiology [11,12].

To date, the majority of studies have predominantly focused on idiopathic PTB. However, perceiving PTB as a single, homogeneous condition is overly simplistic and hinders our ability to recognize distinct classes of pathologies, each with its own unique or overlapping genetic or acquired origins, all ultimately culminating in the same outcome–PTB [13]. A comprehensive evaluation of the specific clinical characteristics of PTB and subgrouping patients based on their clinical presentations before embarking on any genetic study can strengthen the analytical approach, facilitating the identification of distinct causes and genetic markers for subtypes of PTB.

Deciphering the skewed underlying molecular pathways and understanding pathophysiological mechanisms resulting in PTB would eventually provide an opportunity to develop rationale and efficacious targeted intervention strategies to treat PTB etiologically and improving clinical outcomes tailored to the unique phenotypes observed [14–17].

## Cervical shortening—A distinct phenotype in prematurity

Short cervix is defined as a cervical length of ≤25mm before 24 weeks of gestation on transvaginal sonographic examination [18]. Cervical insufficiency is characterized by rapid, painless shortening and opening of the cervix, often accompanied by prolapse of the foetal membranes into the cervical canal, or PPROM, occurring in the second or early third trimester of pregnancy, leading to pregnancy loss or PTB. Isolated (i.e. sporadic) CI occurs in approximately 1–3% of all pregnancies [19], but stands out for its correlation with poor pregnancy outcomes, including PTB rates reaching 40–50%, accompanied by significant perinatal and neonatal mortality rates [20].

Causes of CI are also complex and among others include intra-amniotic infection/inflammation, decidual haemorrhage, uterine overdistension, disruption of maternal-foetal tolerance. There is also evidence of genetics as a contributing factor to CI development [21]. A systematic literature review identified 12 genes linked to CI, majority of which are associated with connective tissue disorders [22]. Indeed, heterogeneous collagenopathies like Ehlers-Danlos (EDS), osteogenesis imperfecta (OI), and Marfan syndromes are clinically recognized risk factors for PTB, PPROM and CI, resulting in a syndromic form of the condition [23], however the direct evidence from genetic studies validating this connection has been notably scarce [24]. This data is further supported by our pilot study in which we conducted clinical exome sequencing in a limited number of patients with CI. Through target gene panel variant analysis and pathway enrichment analysis, we were able to demonstrate that the increased susceptibility to the development of CI is likely influenced by rare damaging variants in genes involved in extracellular matrix (ECM)/collagen synthesis. Therefore, our preliminary study suggests the hypothesis of CI as a subtle form of connective tissue disorder in apparently non-syndromic patients [22].

Considering the aforementioned, the primary objective of the present study is to identify genes and molecular pathways contributing to CI–a clinically distinct group of patients with particularly high risk of PTB. Specifically, our focus is on exploring the hypothesis that CI may stem from rare damaging variants in genes associated with connective tissue and related disorders. To pursue this investigation, we have curated a substantial cohort of females (n = 114) affected by idiopathic CI, capturing a wide range of clinical data (including Beighton [25] and Brighton criteria [26] used to clinically diagnose joint hypermobility and certain connective tissue disorders) that could influence CI development. The exomic data, acquired through next-generation sequencing (NGS) of all patients, has undergone an unbiased and comprehensive characterization. As a result, our study has taken a step toward unravelling the intricate genetics of CI, shedding light on genes not conventionally associated with this condition.

## Methods

### Ethical principles

The research conducted in this study adhered to the ethical guidelines outlined in the Declaration of Helsinki. Approval for the study protocol was obtained from Latvia's Central Medical Ethics Committee under reference number 2/18-03-21 and Lithuanian Kaunas Regional Medical Ethics Committee under reference number 2021-10-14/BE-2-20. All patients were informed of the nature of the study and gave informed consent before enrolment to the study and collection of blood samples.

### Patient inclusion criteria and data collection

Seventy-five patients from Riga Maternity Hospital in Riga, Latvia were included in the prospective longitudinal cohort study. Women with a history of second-trimester pregnancy loss, extreme PTB (22–27 weeks), or those undergoing routine antenatal care were examined for a short cervix or CI and recruited when treatment for cervical shortening was required. The inclusion criteria for these participants were a singleton pregnancy and the presence of a short cervix or CI. Short cervix was defined as a cervical length of ≤25mm before 24 weeks of gestation on transvaginal sonographic examination [18]. Exclusion criteria included multiple pregnancies, uterine anomalies, any genetic disorder other than known connective tissue disorder, age younger than 18 years, and patients in active labour. Active labour was defined as spontaneous preterm labour, characterized by the presence of regular uterine contractions occurring at a frequency of at least four contractions every 20 minutes before 37 weeks of pregnancy. In addition, 32 cases based on the same inclusion/exclusion criteria were recruited from the Lithuanian cohort. Patients were recruited from March 21st 2018 until December 31st 2023. During the same period detailed medical history was extracted from medical records; authors had access to information that could identify individual participants during and after data collection. Additionally, participants also completed a connective tissue dysfunction assessment questionnaire developed for this study (S3 Table). The questionnaire included Beighton criteria for the evaluation of joint hypermobility [25], Brighton criteria for the evaluation of EDS hypermobility type [26], as well as additional questions regarding family history of PTB and connective tissue functionality not covered by the Beighton/Brighton assessment. The Beighton and Brighton criteria are standard tests used to aid in the diagnosis of joint hypermobility syndrome, which may be associated with heritable connective tissue disorders such as Ehlers-EDS, Marfan syndrome, and OI. The Beighton and Brighton criteria were assessed according to the methods described in the original publications [25]. Each question in the connective tissue dysfunction assessment questionnaire was scored as follows: one point was assigned for a positive response (positive phenotype), while the absence of the phenotype was scored as zero points. The questionnaire had a maximum possible score of 25 points, with nine points allocated for the Beighton joint hypermobility criteria and eight points for the Brighton criteria (which also includes Beighton criteria, see S3 Table). Scores for the Beighton and Brighton criteria, as well as total scores, were calculated for each patient individually, as well as for the cohort on average.

During the enrolment process into the study, a clinical assessment of patients was conducted by an obstetrician-gynaecologist. Patients who met the Brighton criteria and those found to have P/LP variants in the connective tissue gene panel were offered a clinical geneticist consultation to evaluate the potential clinical diagnosis of a connective tissue disorder.

Controls for this study were chosen from an in-house WES/whole genome sequencing database (n = 588; 34.8% females and 65.2% males) comprising individuals who underwent genetic testing for various disorders, primarily neurologic or cardiological conditions. The

presence of CI in the controls was not assessed. However, given the anticipated rarity of CI in the population, it is not expected to markedly influence genetic association analysis.

## DNA sample processing and bioinformatics pipeline

The exomes of all CI patients were analysed using the Twist Human Core Exome Kit (Twist Bioscience, San Francisco, CA, USA). Sequencing was performed on the NextSeq 500 platform (Illumina) with 75 paired-end (PE) reads. This sequencing service was provided by CeGaT GmbH in Tübingen, Germany.

**Variant calling and Quality Control (QC).**   Bioinformatics analysis was conducted, and all project samples and associated data were securely stored on the DNAnexus cloud platform. Utilizing an in-house pipeline implemented within the DNAnexus cloud platform, WES data from samples related to patient (CI) and control samples, provided in FASTQ format, were aligned to the GRCh38 reference genome using BWA-MEM [27]. Duplicate reads were systematically removed from the analysis. Variant calling for each sample was performed using DeepVariant (v1.2.0) [28], resulting in variant calls provided in both VCF and GVCF formats. To ensure data quality, reference metrics for quality control (QC) were calculated based on an in-house WES database, and these metrics were subsequently applied as QC criteria to all CI sample data. All CI WES sample data successfully met these QC criteria and were considered for further analysis. For joint variant calling, GVCF format files from both CI samples and controls were integrated using GLnexus (v1.4.1) to generate the final variant calling output in pVCF format [3].

**Removal of relatives.**   Kinship analysis was conducted using the PLINK 1.9 software suite with a relationship-based pruning approach (—rel-cutoff 0.1) (Purcell, Neale et al. 2007). This analysis aimed to identify and remove individuals with relationships up to the 3rd degree. In cases where related pairs were detected, one individual from each pair was systematically excluded from subsequent analysis. As a result, no cases and 16 control individuals were excluded from the final dataset.

**Variant filtering and prioritization.**   Variants were subjected to annotation using Ensembl VEP (release 107) [29]. Subsequent variant exclusion was carried out through the application of bcftools (v.1.14) [30] and vcftools (v0.1.16) [31], adhering to the following filtering criteria:

1. Variants with a read depth (DP) of less than 10 were excluded.

2. Heterozygous variants with a variant allele frequency (VAF) of less than 0.3 were removed.

3. Variants failing the Hardy-Weinberg equilibrium (HWE) test with a p-value less than 1.5e-15 were excluded.

4. Variants with call rates in less than 99.9% of all individuals were eliminated.

5. Variants with an allele count exceeding 5% of the entire cohort (both cases and controls) were removed.

6. Variants with an allele frequency (AF) in the gnomAD Non-Finnish European population (NFE) or gnomADg NFE exceeding 0.01 were also excluded.

The filtering pipeline also incorporated an individual-level variant missingness filter using vcftools (v0.1.16) (—missing-indv) to exclude individuals with excessive amount of missing genotype data. No individuals had a missingness level exceeding 10%.

Loss-of-function (LOF) variants were identified utilizing the Ensembl VEP LOFTEE [32] plugin. Specifically, variants with VEP consequences such as stop gained, frameshift variant,

splice acceptor variant, and splice donor variant were included. Variants marked with a LOF-TEE filter "LC," indicating low-confidence LOF variants that failed at least one LOFTEE filter, were excluded from the analysis. Additionally, variants with VEP consequences of splice region and synonymous, accompanied by a SpliceAI [33] score exceeding 0.5, were incorporated into the LOF variant set.

Missense variants were selected based on a PHRED-scaled CADD [34] score greater than 25 and a REVEL [35] score exceeding 0.65. Only variants with VEP consequences such as missense variant, in frame deletion, in frame insertion, and start lost were included in the missense variant set.

## Genome wide rare variant burden test

We conducted three gene-based exome-wide variant association tests: SKAT, SKAT-O, and burden test using the default settings of the SKAT R package v.2.2.4 [36]. This package allows us to treat genes as the fundamental units for statistical testing, enhancing our ability to identify disease-associated genes. The SKAT test is powerful when a small fraction of variants is causal and effects are mixed. Burden tests are sensitive when a significant fraction of variants is causal with consistent effects. The SKAT-O test combines strategies from both to maximize power. All three tests were applied to three variant sets: LOF variants, missense variants, and a combination of LOF and missense variants. CI samples were matched against 527 controls (including 62 whole genome sequencing samples). We converted the filtered variant file from VCF to plink format and used the SKAT R package for analysis.

## Individual variant analysis in SeqR platform

Simultaneously with the joint analysis, individual variant analysis of genes clinically associated with connective tissue disorders were conducted for each case using a locally installed SeqR platform [37]. For each individual case, mitochondrial variants were identified using Strelka2 [38] with the—callContinuousVf chrM option. The VCF variant calls from Strelka2 were merged with VCF DeepVariant, resulting in the final VCF variant call file. This file was subsequently annotated using Ensembl VEP, and only variants with a 'PASS' annotation were uploaded to the SeqR platform for further analysis.

Following variants were retained in the SeqR platform: nonsense, essential splice site (applying SpliceAI cut off of 0.2), missense and in-frame indels. An allele frequency (AF) threshold of 0.001 was applied for variants occurring in gnomAD exomes, gnomAD genomes, or the TOP-Med population database. Custom filters were overridden if a variant was found in ClinVar as P/LP or in the HGMD database as (likely) disease-causing.

We compiled a connective disorder gene panel by accessing data from the PanelApp database's homepage (https://panelapp.genomicsengland.co.uk/), encompassing genes from EDS, OI, and Stickler syndrome panels (n = 102, S4 Table). Specifically, we included genes classified under green (established) and amber (putative) evidence categories, omitting those marked as red. We focused particularly on these genes as particular disorders are clinically associated with CI, and this is also in accordance with our hypothesis that CI is a subtle form of collagenopathy. Although due to allelic heterogeneity and pleiotropy analysis of this gene panel could reveal non-typical connective tissue disorders. Current gene panel also covered all genes previously linked to CI as identified in our pilot study [22]. Subsequently, to confirm any existing connective tissue disorder at the molecular level, the variants returned from this panel underwent classification according to ACGS guidelines [39] by two certified molecular geneticists.

Next, we correlated the variants identified in connective tissue gene panel among our patients with their Beighton/Brighton scores and responses to the connective tissue

dysfunction assessment questionnaire (S3 Table). Given that the phenotype of isolated (non-syndromic) CI lacks a clear-cut gene-disease association in a Mendelian fashion, no SNV can be evaluated using ACGS criteria as it was done in the previous step. To overcome this, we implemented a cut-off of AF<0.001 in population databases and considered variants with a CADD score >10, designating these variants of interest as 'rare damaging SNVs'.

### Gene expression data

For the gene expression data, we sourced information from the consensus dataset on proteinatlas.org, which combines the HPA and GTEx transcriptomics datasets, and is presented in normalized expression levels (TPM).

### Pathway-based burden test of ECM-related pathways

Pathway-based burden test was performed for the ECM pathways. Specific ECM pathways were taken from our pilot study [22] and encompassed following pathways: collagen formation, ECM-receptor interaction-Homo Sapiens (human), ECM organization, collagen biosynthesis and modifying enzymes, type I hemidesmosome assembly, assembly of collagen fibrils and other multimeric structures, laminin interactions, collagen chain trimerization, integrin, non-integrin membrane-ECM interactions, ECM proteoglycans, beta 1 integrin cell surface interactions, focal adhesion-Homo Sapiens (human), alpha6 beta4 integrin-ligand interactions. Additionally, we assessed 'HSP90 chaperone cycle for steroid hormone receptors (SHR) in the presence of ligand' pathway. The test was conducted against 527 (including 62 whole genome sequencing samples) controls in a manner identical to the gene-based rare variant burden test, but based on rare damaging variants in a pathway. Each test (Burden, SKAT, SKAT-O) was performed separately for LOF and missense variants, as well as jointly for LOF and missense variants.

### Pathway analysis

In order to perform genome wide pathway enrichment of genes having rare damaging variants as well as to obtain insight about genes shown to be enriched for rare damaging variants, we conducted pathway enrichment analysis using ConsensusPathDB (available at http://cpdb.molgen.mpg.de/CPDB). We exploited interaction database gene set analysis function 'over-representation analysis' and looked for 'Pathway-based sets' in all built-in pathway databases with a p-value cut-off of 0.01. Genome-wide pathway enrichment analysis was conducted using two distinct modes: initially, genes were selected based on SNVs present exclusively in the CI group (genes shared between cases and controls). The second pathway analysis mode involved selecting genes devoid of rare damaging SNVs in controls (with no gene overlap between cases and controls).

## Results

We conducted whole exome sequencing (WES) analysis on 114 patients diagnosed with a short cervix, CI or a history of CI in previous pregnancies. Patients were recruited from Latvia (n = 82) and Lithuania (n = 32)–genetically related populations of the Baltic region [40,41]. Average age of our patients was 33 ± 5.4 years; 43 patients were primiparas and 71 multiparas. On average the short cervix was diagnosed on 20 ± 4 week of gestation with average cervical length at the time of diagnosis being 17 mm. The phenotype data obtained from the connective tissue dysfunction assessment questionnaire (S3 Table), developed by our group for this study and incorporating both the Beighton and Brighton criteria [25,42], was subjected to

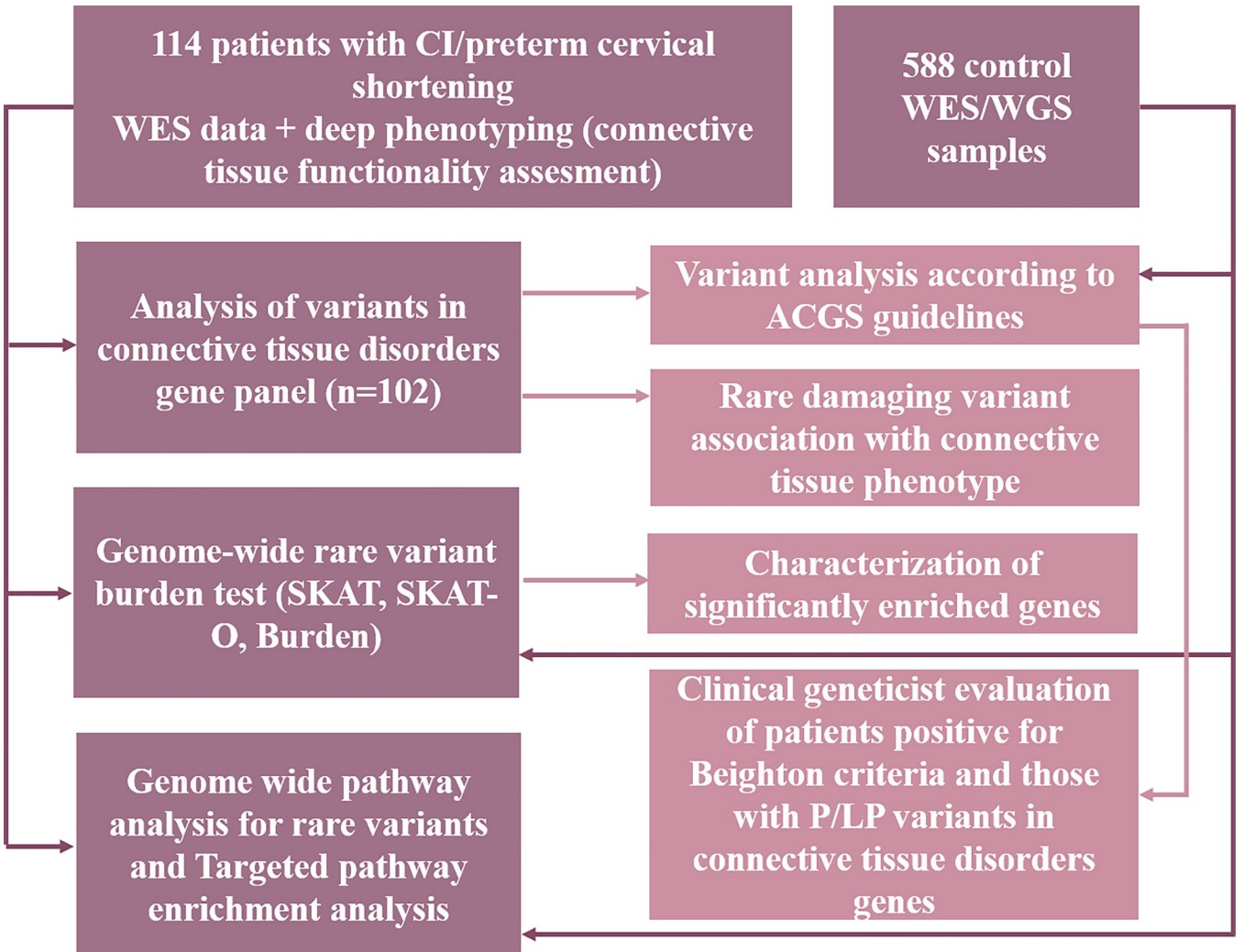

**Fig 1. Overview of the study design.** CI–cervical insufficiency. WES–whole exome sequencing. WGS–whole genome sequencing. ACGS–The association for clinical genomic science. P/LP–pathogenic/likely pathogenic variant.

correlation analysis with the genetic findings. The WES data from all patients fulfilled the requisite quality criteria, ensuring its suitability for subsequent analysis. An overview of the study design is depicted in the Fig 1. As controls, we utilized data from 588 individuals from our internal in-house WES database.

## Analysis of variants in genes implicated in connective tissue disorders

Out of 166 filtered single nucleotide variants (SNVs) within genes known to cause connective tissue disorders in CI group (n = 114), four variants were classified as P/LP (Table 1) according to ACGS guidelines [39]. From the control samples (n = 588) 763 variants remained after filtering step, and 39 variants were classified as P/LP (S1 Table), including *FKBP14* variant NM_017946.3:c.362dup which was also found in CI group. Thus, frequency of P/LP variants in connective tissue gene panel was 0.035 in patient and 0.067 in control group (p = 0.193). Clinical geneticist evaluations of patients carrying P/LP variants did not reveal any underlying Mendelian disorder during consultation (Table 1).

**Table 1. Pathogenic / Likely pathogenic variants in connective tissue and related disorders gene panel and anamnesis of patients with cervical insufficiency.**

| Age | Beighton/ Brighton/ Total score | Gene Associated Disease | Coding sequence Protein alteration ClinVar ID | ACGS Criteria | RNA expression within cervix (TPM) | Medical history | Clinical geneticist conclusion | CI diagnosis week, cervical length | Pregnancy outcome |
|---|---|---|---|---|---|---|---|---|---|
| 38 | 3 / 1 / 7 | *COL1A1* AD Ehlers-Danlos syndrome, arthrochalasia type, AD osteogenesis imperfecta | NM_000088.3: c.1663C>T p.(Pro555Ser) ClinVar ID: 1702168 | LP (PM1, PP3, PP2, PM5 [43]) | 1284.0 | Urinary incontinence, scoliosis, umbilical hernia. One PTB at 23 weeks of gestation (intrauterine foetal death), four pregnancy losses, two term deliveries | The patient could not attend clinical geneticist consultation | 16th week of gestation,17 mm | Term delivery |
| 33 | 0 / 1 / 1 | *COL6A1* AD/AR Bethlem myopathy, AD/AR Ullrich congenital muscular dystrophy | NM_001848.2: c.244C>T p.(Arg82Ter) ClinVar ID: 1375454 | LP (PVS1, PM2) | 483.6 | Excessively thin /elastic skin, abnormal scarring. One term delivery, one PTB (23rd week of gestation) | Does not confirm *COL6A1*-associated disease. Due to signs of muscular hypotonia, the patient has been sent for a more in-depth neurological examination | 20th week of gestation,0 mm | Extremely preterm (22–27 weeks of gestation) delivery |
| 34 | 0 / 0 / 0 | *FKBP14* AR Ehlers-Danlos syndrome, kyphoscoliotic type, 2 | NM_017946.3: c.362dup p. (Glu122ArgfsTer7) ClinVar ID: 279809 | LP (PVS1, PM3) | 9.5 | Three early pregnancy losses, one PTB (weeks 22–27) | Does not confirm *FKBP14*-associated disease | 22nd week of gestation,0 mm | Extremely preterm (22–27 weeks of gestation) delivery |
| 39 | 4 / 1 / 6 | *ALPL* AD/AR odonto-hypophosphatasia | NM_000478.6: c.571G>A p.(Glu191Lys) ClinVar ID: 13670 | P (PS3, PS4, PM1, PP3) | 3.3 | Five pregnancy losses, one PTB (31 week of gestation), family history positive for PTB | At the age of 20, six teeth were lost. Although there are no persuasive features of hypophosphatasia, additional analyses, including alkaline phosphatase, Ca, P in blood, and osteodensitometry have been requested | 15th week of gestation,11 mm | Very preterm (28–31 weeks of gestation) delivery |

*LP–likely pathogenic; P–pathogenic. ACGS criteria according to [39]. Beighton criteria according to [25]. Brighton criteria according to [26]. RNA expression obtained in GTEx transcriptomics dataset.*

Next, we correlated the variants identified in connective tissue gene panel among our patients with their Beighton/Brighton scores and responses to the connective tissue dysfunction assessment questionnaire (S3 Table). The analysis revealed a total of 132 rare damaging SNVs in 64 genes known to be clinically associated with connective tissue disorders in 77 patients (67.5%). All genes with rare damaging SNVs showed at least minimal expression within cervical tissues (Fig 2A). The average number of rare damaging SNVs per patient with a variant was 1.71, compared to 1.79 identified in controls (p = 0.474).

Overall 27 (23.7%) patients had joint hypermobility according to Beighton score and 12 (10.5%) were positive for Brighton criteria (Fig 2B). Average age of patients with joint hypermobility comparing to those not showing signs of hypermobility according to Beighton was 31.8 ± 5.3 and 32.8 ± 5.1 (p = 0.4) (Fig 2C).

Among the 77 patients who had at least one rare damaging variant, the scores of Beighton/ Brighton/connective tissue dysfunction assessment questionnaire were 1.98/1.21/4.27,

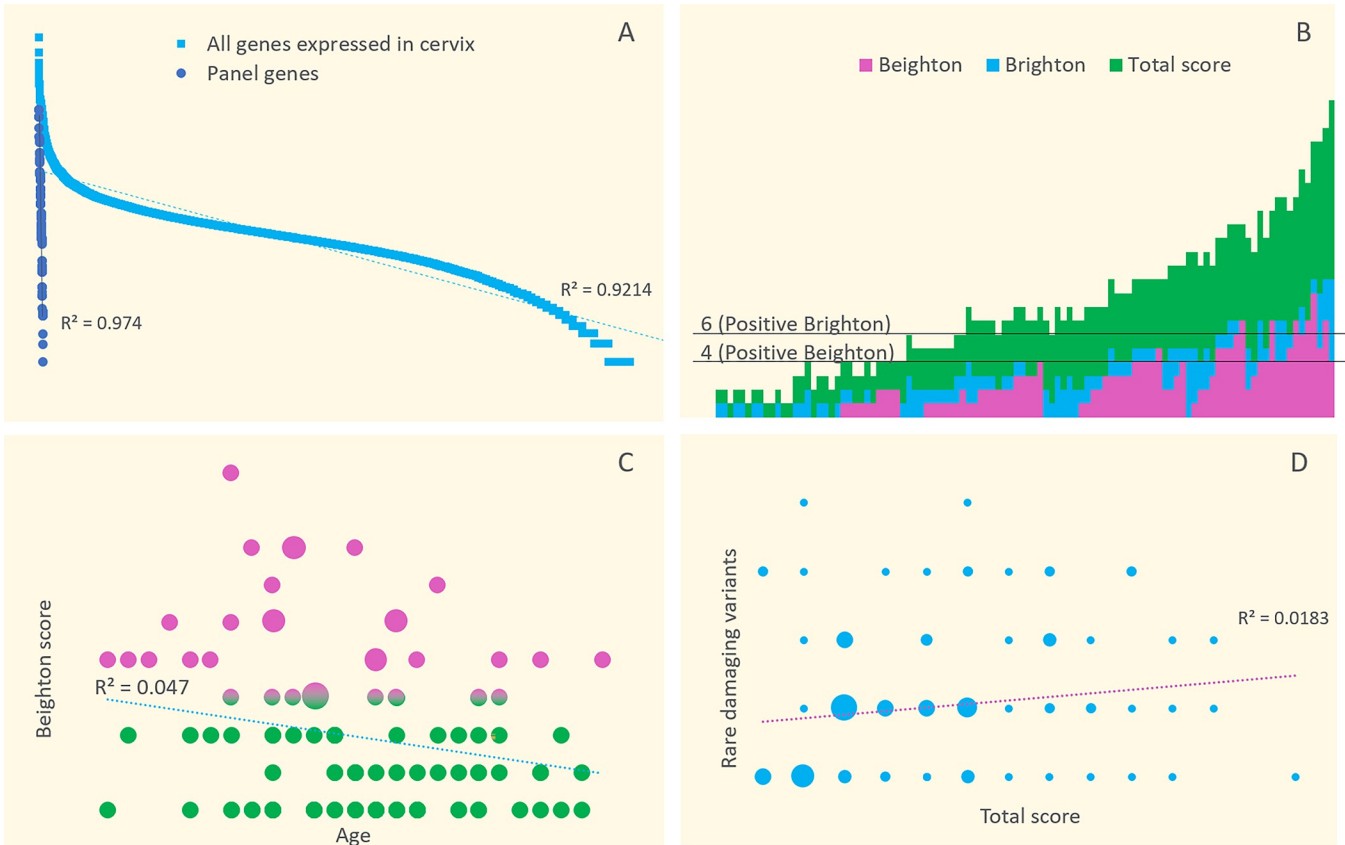

**Fig 2. Exploring genetics of cervical insufficiency: Connective tissue perspective.** A) Expression (Y axis) of connective tissue disorder genes (blue) in relation to all genes showing at least some expression within cervix (TPM>0) (light blue) (X axis). B) Beighton (pink) / Brighton (blue) / Total score (green) (Y axis) obtained by each study patient (X axis) in the connective tissue disfunction assessment questionnaire. C) Patients age (X-axis) correlation with Beighton hypermobility score (Y-axis). A larger circle size indicates a greater number of patients with the same age and Beighton score. D) Total score obtained in connective tissue dysfunction assessment questionnaire (X-axis) correlation with number of rare damaging gene variants in connective tissue gene panel (Y-axis). A larger circle size indicates a greater number of patients with the same score and number of rare variants.

respectively. On the other hand, the 37 patients with no SNVs identified had corresponding scores of 2.0/1.19/4.29 (p > 0.05). Conversely, patients with joint hypermobility according to the Beighton criteria ($\geq$ 4 points) or a positive Brighton score had, 1.24/1.45 rare damaging SNVs accordingly vs 1.11/1.16 rare damaging SNVs of those without joint hypermobility according to Beighton (< 4 points; 87 patients) or a negative Brighton score (102 patients) (p > 0.05).

Individual number of rare damaging SNVs showed only a trend towards a positive correlation with the total score obtained in connective tissue dysfunction assessment questionnaire without reaching a statistical significance (Fig 2D).

## Genome-wide rare variant burden test results

To identify genes associated with CI without a priori hypothesis (i.e. genome wide approach), we conducted three rare damaging SNV burden tests. Each test identified a number of genes enriched with rare damaging SNVs (nominal p-value < 0.01): LOF (loss of function) burden test n = 27 genes; LOF SKAT-O n = 33; LOF SKAT n = 15; missense burden n = 112; missense SKAT-O n = 123; missense SKAT n = 54; LOF/missense burden n = 122; LOF/missense SKAT-O n = 136; LOF/missense SKAT = 64 genes. In total 179 unique genes were found in at

**Table 2. Top-20 pathway enrichment results of genes having burden of rare damaging SNVs in patients with cervical insufficiency.**

| Pathway name | p-value | Pathway name | p-value |
|---|---|---|---|
| Prednisolone Action/Metabolism Pathway | 8.73E-05 | Signalling by activated point mutants of FGFR1 | 0.00454 |
| HSP90AA1 | 0.000817 | PI3K-Akt signalling pathway | 0.00463 |
| Corticosteroids and cardio protection | 0.002 | Nuclear receptors | 0.00535 |
| Recycling of eIF2:GDP | 0.00236 | FGFR1c ligand binding and activation | 0.00542 |
| VEGFR1 specific signals | 0.00246 | Tetrahydrobiopterin (BH4) synthesis, recycling, salvage and regulation | 0.00542 |
| Transport of inorganic cations/anions and amino acids/oligopeptides | 0.0027 | Extracellular matrix organization | 0.00563 |
| Tyrosine metabolism | 0.00346 | PI3K-Akt signalling pathway—Homo sapiens (human) | 0.006 |
| Collagen biosynthesis and modifying enzymes | 0.00359 | Ras signalling pathway—Homo sapiens (human) | 0.00622 |
| Organic anion transporters | 0.00374 | FGFR2c ligand binding and activation | 0.00636 |
| ROS and RNS production in phagocytes | 0.00424 | eNOS activation | 0.00636 |

least one of the tested models (S2 Table). To gain insights into the relevance of these enriched genes in the context of CI, we conducted an evaluation that encompassed pathway enrichment analysis (Table 2), gene expression data as well as clinical and functional association (S2 Table).

Thus, out of 179 genes enriched for rare damaging SNVs, 159 had at least some expression within cervical tissues (TPM $\geq$ 0.1); 121 genes did not show any clinically associated phenotype (OMIM, PanelApp). Fifty-eight genes had known association with certain Mendelian diseases, although vast majority of these are unrelated to the phenotype of interest (i.e. connective tissue and related disorders).

A closer examination of genes exhibiting an increased variant burden and pathway enrichment results has sparked particular interest in the *PGR* (progesterone receptor), *HSP90AA1* (heat shock protein HSP 90-alpha) and *NR3C1* (glucocorticoid receptor) genes.

Progesterone as a naturally occurring steroid is necessary for the maintenance of pregnancy and plays a key role in maintaining cervical integrity prior to labour induction [44]. *PGR's* expression within cervical tissue is 61.6 TPM. Six out of 114 patients with CI had rare SNVs in *PGR* (Fig 3) without having any additional rare damaging SNVs in the connective tissue disorder gene panel. Three variants were localized in (highly) missense variation intolerant regions (assessed through Metadome) [45] within known functional progesterone receptor domains– DNA binding domain and ligand binding domain. Across 588 control samples we identified four *PGR* variants in six patients, two SNVs were present both in patients and controls. Thus, frequency of *PGR* SNVs in patients was 0.052 vs 0.011 in controls (p = 0.0034). 3D analysis of the identified variants demonstrated that position Arg615 is localized proximally to the DNA binding site and amino acid change potentially could affect receptor's and DNA interaction affinity. In turn, Arg788 localizes in the progesterone binding site and its change to Trp is predicted to affect ligand binding, potentially leading to the protein loss of function. Three remaining SNVs were predicted as benign by in silico tools, two of which are resided in an N-terminal (modulatory) domain and one SNV–C-terminally out of any domain.

Next, across our patients we identified two rare damaging *HSP90AA1* variants: NM_001017963.2:c.1843T>C p.(Tyr615His) and c.1724C>G p.(Ser575Cys) localized in a highly conserved and missense constraint positions. In controls we identified two *HSP90AA1* variants, including c.2536_2538del located C-terminally out of domain and c.942A>C located in a missense intolerant region (p = 0.0001). *HSP90AA1* has particularly high expression within cervix– 272.1 TPM.

Two of our patients had two *NR3C1* SNVs: NM_001018077.1:c.1084G>A p.(Glu362Lys) and c.1475A>G p.(Lys492Arg). A more in-depth analysis of variants demonstrates that the

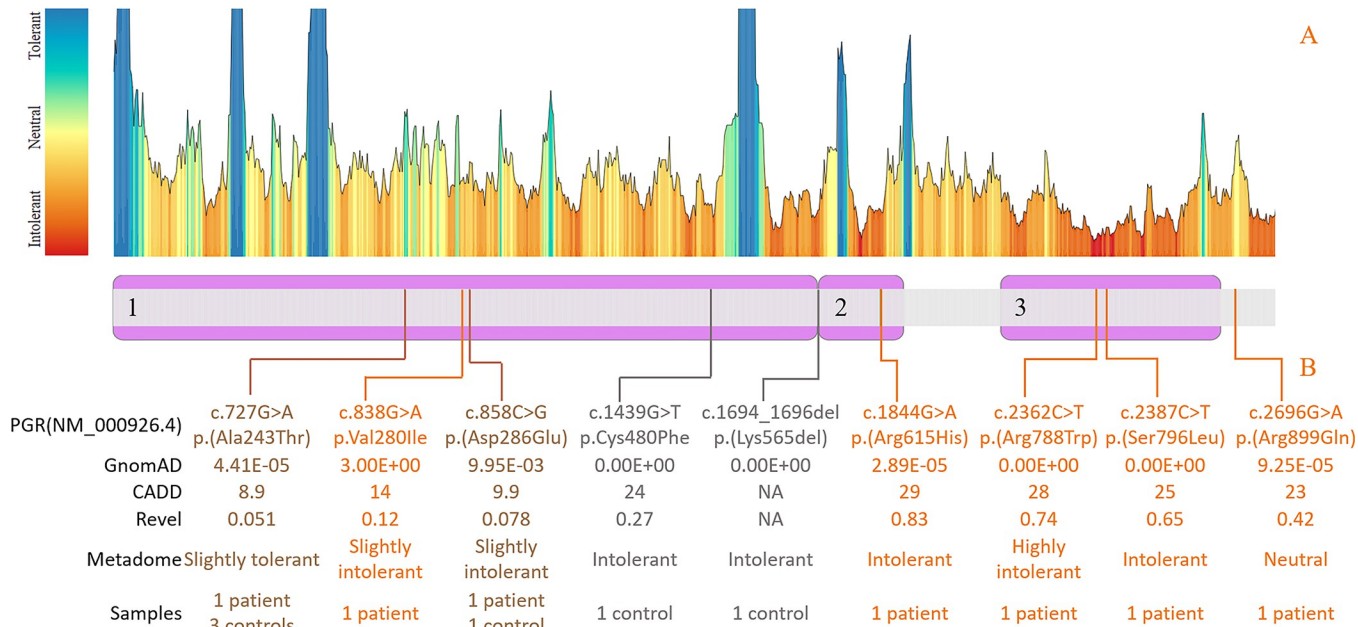

**Fig 3. *PGR* (progesterone receptor) variants identified in patients with cervical insufficiency and *PGR* amino acid positions tolerance map to missense variation.** PGR functional domains: 1) an N-terminal (modulatory) domain. 2) a DNA binding domain. 3) a hormone/ligand binding domain. GnomAD–variant frequency in a GnomAD V4 population database. CADD–combined annotation dependent depletion score, variant effect in silico prediction tool. Revel score–variant effect in silico prediction tool.

position Lys492 plays a role in DNA binding and undergoes modification to N6-acetyl-lysine. Substituting the position with arginine could potentially impact the glucocorticoid receptor's DNA binding capability. In turn, Glu362 is located within the disordered region. Across 588 controls we encountered three *NR3C1* variants, including c.1094A>G in one sample located in a missense tolerated region and c.1639G>T located in missense intolerant region out of domain in two samples (p = 0.0007).

## Pathway enrichment analysis results

Next, we performed targeted pathway-based burden tests (Burden, SKAT, SKAT-O) for the pathways (n = 14) demonstrating enrichment in our pilot study [22] (Methods). Additionally, we analysed burden of rare damaging variants in 'HSP90 chaperone cycle for steroid hormone receptors in the presence of ligand' pathway which includes *PGR*, *HSP90AA1* and *NR3C1* genes. Result revealed rare deleterious variant enrichment (nominal p-value, p < 0.05) in a few ECM-related pathways by some of the enrichment tests used: missense SKAT test revealed enrichment within 'focal adhesion' pathway; LOF SKAT-O and burden tests identified enrichment in 'collagen formation', 'collagen chain trimerization', and 'collagen chain trimerization pathway'.

All tests used demonstrated statistically significant rare damaging variant enrichment in steroid hormone receptors pathway containing *PGR*, *HSP90AA1* and *NR3C1* genes, even if corrected for Benjamini-Hochberg method, e.g., LOF/missense burden test, p-adj = 5.37E-06.

In turn, genome-wide pathway enrichment analysis uncovered predominant enrichment in ECM pathways as well as highlighted several distinct pathways depending on the analysis mode (Fig 4A and 4B).

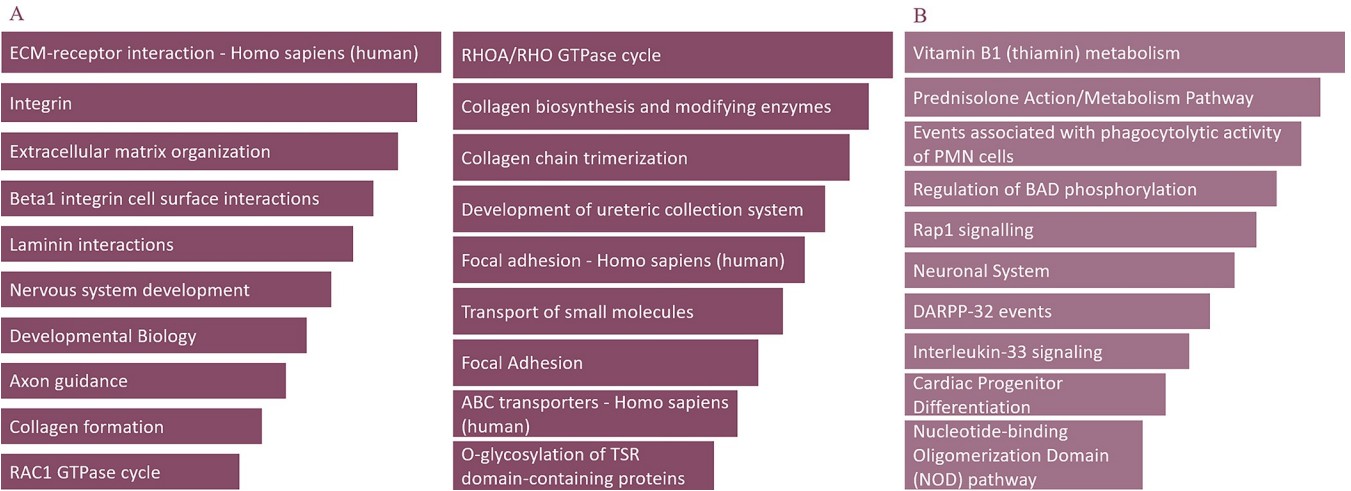

**Fig 4. Genome wide pathway enrichment analysis results.** A. Genome-wide pathway enrichment conducted by selecting genes with single nucleotide variants present exclusively in the cervical insufficiency group (genes shared between cases and controls). B. Genome-wide pathway enrichment conducted by selecting genes devoid of rare damaging single nucleotide variants in controls (with no gene overlap between cases and controls).

## Discussion

Our overarching objective was to delve into the genetic aetiology of CI. The impetus for our study emerged from a critical clinical imperative, given the current limitations in timely prediction and prevention of CI consequences in clinical settings [46]. To accomplish this, we performed WES on a meticulously phenotyped cohort comprising individuals with preterm cervical shortening or CI. Employing a comprehensive analytical approach, we ensured a thorough exploration of the WES data. Our patient cohort, consisting of 114 females from Latvia and Lithuania, represents the most extensive group of CI patients subjected to WES to date. This underscores the novelty and significance of our study in shedding light on the genetic landscape of CI.

### Rethinking the role of connective tissue gene variants

Several connective tissue disorders are well-established in their clinical links to adverse pregnancy outcomes such as PTB, PPROM, and CI, [23,47]. Supporting this, compelling studies shed light on the intricate connection between CI and predisposition to pelvic organ prolapse and uterine rupture attributing to impaired collagen functioning [48]. Despite the long-standing implication of collagen in the development of CI, direct evidence from genetic studies validating this connection has been notably scarce [24]. Findings in our pilot study on the genetics of CI [22], prompted the formulation of a hypothesis that CI is a subtle manifestation of connective tissue disorder. To rigorously explore and substantiate our hypothesis, our current investigative approach comprised three key components. Firstly, we assessed the connective tissue functionality of individuals with CI by employing recognized Beighton and Brighton scores clinically used to diagnose joint hypermobility and some connective tissue disorders complemented with targeted inquiries concerning personal and family histories of PTB, CI, and general connective tissue functionality (S3 Table). Secondly, we conducted a comprehensive analysis of variants within a connective tissue gene panel. To contextualize the frequency of P/LP variants in collagen genes, we juxtaposed our findings with an analysis of our internal WES database, representative of our population (n = 588). Lastly, clinical geneticist consultations were extended to individuals exhibiting positive Brighton criteria and those harbouring

P/LP variants in the analysed gene panel to assess whether a clinical diagnosis of connective tissue disorder can be established.

Analysis of genes causing connective tissue disorders according to ACGS guidelines [39] revealed four P/LP variants in *COL1A1*, *COL6A1*, *FKBP14*, and *ALPL*. *COL1A1* is associated with several conditions, including arthrochalasia-type EDS and OI. It is considered that the penetrance in individuals heterozygous for a *COL1A1* pathogenic variants is 100%, although disease expression may vary considerably, even within the same family [49]. Furthermore, *COL1A1* has already been implicated in the development of CI from the data of case-control studies [50,51]. Regrettably, the patient harbouring the LP variant in *COL1A1* in our cohort proved unreachable for discussion regarding her genetic testing outcomes. Absent a clinical geneticist's evaluation, asserting whether patient clinically manifests a diagnosis related to *COL1A1*-associated conditions remains challenging. There are no reports linking *COL6A1*, *FKBP14*, and *ALPL* associated disorders with CI. Patients carrying P/LP variants in these genes did not demonstrate clinical features associated with the particular gene disorders.

Thus, applying ACGS guidelines to assess P/LP variants of connective tissue gene panel, indicated their rarity in patients with CI. No variants could be associated with monogenic connective tissue disorder clinically associated with CI (except the case with *COL1A1* that could not be resolved). This suggests that CI in non-syndromic patients is not attributed to a single gene in a Mendelian fashion.

To gain further insights into the potential relationship among the three studied entities–genetics, CI, and connective tissue disorders–we correlated rare SNVs of connective tissue genes with the phenotypic data of patients with CI. We developed connective tissue dysfunction assessment questionnaire (S3 Table), aiming to encompass a comprehensive range of phenotypic information related to connective tissue functionality. For instance, we expanded the validated Beighton/Brighton questionnaire by incorporating details about muscle pain, a common symptom of connective tissue disorders [52]. Similarly, questions pertaining to scoliosis and the occurrence of frequent or atypical bone fractures were included to capture subtler phenotypical expressions in seemingly unaffected individuals. Recent studies indicate that women with a history of CI experienced a higher rate of pelvic organ prolapse and urinary symptoms [53], therefore, this information was also incorporated into the questionnaire.

Analysing connective tissue disorder related phenotypes revealed that in our cohort, 23.7% of patients exhibited joint hypermobility based on the Beighton criteria; while 10.5% met the clinical criteria outlined in the Brighton scoring system, theoretically indicative of hypermobility spectrum disorder. Correlation analysis between number of rare damaging SNVs and connective tissue phenotypes demonstrated a positive trend but lacked a clear correlation, hinting at the multifactorial nature of CI development. The clinical geneticist' conclusion, following the evaluation of these patients, could only confirm benign joint hypermobility spectrum. Additional features, as assessed using the Brighton criteria, were considered as separate phenotypic units, not confirming any disease attributable to classic connective tissue disorder. As the literature suggests, joint hypermobility, including hypermobility type EDS–the most prevalent hypermobility spectrum disorder, currently lacks an identified genetic cause [42]. Notably, none of the patients positive for the Brighton criteria in our cohort had a P/LP variant in the connective tissue disorder gene panel either. Hence, the question of whether CI lies on a spectrum of joint hypermobility attributable to sub-clinically reduced connective tissue function remains open and requires further exploration.

Despite the negative results of the rare variant analysis of connective tissue genes, targeted and genome-wide examination of molecular pathways revealed a significant enrichment of variants in genes associated with collagen pathways. Interestingly, ECM-associated pathways were predominantly enriched when analysing genes with excluded overlapping variants in

controls. The disparity in pathway outputs between the two analysis modes suggests that genotype-phenotype interactions may be influenced by diverse variant effects, potentially involving variants that either protect against or contribute to the CI phenotype. Pathway analysis findings not only replicate several of ECM-associated pathways identified as enriched in our pilot study [22], but also further strengthen the indication of potential cumulative involvement of multiple inherited connective tissue gene variants in the development of CI as a multifactorial disorder.

### Rare variant burden analysis identifies *PGR* as a promising gene for CI

In order to continue a comprehensive search of a genetic markers with the potential role in the development of CI, we performed a genome wide burden analysis of rare damaging variants. As a result, 179 genes occurred to be significantly enriched as demonstrated three different tests used. Our attention captured involvement of genes associated with steroid pathways, specifically *PGR*, *HSP90AA1* and *NR3C1*. *PGR* encodes progesterone receptor which mediates the physiological effects of progesterone. Both cervical stromal and epithelial cells express progesterone receptors [54]. Progesterone is known for its role in maintaining pregnancy by preserving uterine quiescence in the latter half of pregnancy, limiting the production of stimulatory prostaglandins, and inhibiting the expression of contraction-associated protein genes within the myometrium. It appears that progesterone also inhibits cervical collagen decomposition [55]. The onset of labour, both at term and preterm, is associated with a functional withdrawal of progesterone activity at the level of the uterus. Progesterone was the first FDA approved supplementation for the prevention of PTB [56] and evidence of the therapeutic utility of progesterone for the prevention of CI and PTB in women at-risk is well documented [57]. In our study, we identified a statistically significant prevalence of rare damaging *PGR* variants in cases compared to controls. *In silico* and 3D analysis of the identified variants demonstrated that three variants, one localized within the DNA binding domain and two within the hormone/ligand binding domain, predict variants as pathogenic. In theory, such variants could lead to the reduction of activity (or even loss of function) of the progesterone receptor via an inability/reduced ability to be stimulated by progesterone or an inability/reduced ability to bind the DNA. We speculate that impaired PGR functioning due to pathogenic variants could eventually lead to preterm cervical shortening by two possible pathways: 1) malfunctioning endogenous progesterone effect leading to an abrupt progesterone withdrawal; 2) unsuccessful exogenous progesterone administration. In fact, progesterone therapy is effective only in a subset of patients [58], leaving the rest without a clear understanding of the underlying aetiology. In theory, *PGR* variants may thus account for a proportion of unsuccessful progesterone administration.

Next, *NR3C1* encodes the prednisolone receptor, a steroid hormone receptor whose endogenous agonist is cortisol. This receptor regulates the expression of anti-inflammatory and immunosuppressive effects. Importantly that progesterone, which shows structural similarities to glucocorticoids, can bind the intracellular glucocorticoid receptor, promoting maternal immune tolerance to foetal alloantigens through a wealth of immunomodulatory mechanisms [59]. Literature suggests that variants in *NR3C1* are associated with the AD glucocorticoid resistance characterized by impaired cortisol signalling, clinically resulting in hypoglycaemia, hypertension, metabolic alkalosis, chronic fatigue and female infertility [60]. In turn, heat shock protein 90 (*HSP90AA1*) assists in folding the prednisolone receptor and facilitates its transport into the nucleus [61]. Both genes exhibit significant expression within cervical tissues. Once again, we may cautiously assume that variants in these genes could potentially disrupt the normal prednisolone action/metabolism pathway leading to increased susceptibility

to infections or compromising the immunosuppressive effects crucial for maintaining maternal immune system tolerance during pregnancy eventually resulting in preterm cervical shortening and/or PTB.

Notably, rare damaging variants in the steroid hormone receptors pathway reached exome-wide significance in our moderate sample size cohort, pointing to the significance of these variants for the pathophysiology of CI. In a recent study, Wang et al. also discovered a variant burden in genes regulated by the PGR, proposing it as a predictor of responses to progestin treatment for PTB [62]. Hence our findings for the first time corroborate Wang et al.'s suggestion that rare damaging SNVs within the PGR pathway may play a significant role in PTB development.

While our findings suggest intriguing associations between rare damaging variants in *PGR*, *HSP90AA1*, *NR3C1* and CI, it is crucial to approach these observations with caution. These are early assumptions that require rigorous validation through further investigations. It is noteworthy that standard variant interpretation guidelines do not assess variants in genes lacking interactions with clinical phenotype [63]. Functional studies of the identified variants are necessary to elucidate their specific impact on the progesterone and prednisolone pathways and, by extension, their potential contribution to CI.

## Study limitations

While our study represents the largest cohort of comprehensively phenotyped females with CI undergoing WES, it is important to acknowledge the limitations that may impact the generalizability of our findings. The relatively modest sample size, though substantial, is still insufficient to definitively ascertain the spectrum of possible connective tissue disorders resulting from P/LP variants in connective tissue or other genes. Establishing a more comprehensive understanding of the causative role of rare damaging variants, particularly in influencing the complex phenotype of CI and the broader spectrum of connective tissue disorders, necessitates a larger cohort. It is noteworthy that our study relied on control data from our internal WES database, primarily composed of individuals undergoing testing for cardiac and neurological diseases. This targeted selection may introduce a bias, limiting the generalizability of our results to a broader population. Moreover, our study is constrained by the absence of a matched control group that underwent thorough phenotyping in a manner analogous to the study cohort. A meticulously phenotyped control group would have provided a crucial baseline for estimating the prevalence of hypermobility within the CI group.

## Concluding remarks and the future perspectives

Number of genes identified as potentially contributing to the CI/PTB are limited [64,65]. The current understanding of the pathophysiology of CI primarily revolves around inflammation and connective tissue dysfunction posing a challenge to unravelling novel pathways and genes contributing to the pathobiology of this condition. Our study provides evidence that unexplored avenues exist. While our primary goal was to elucidate the relationship between CI and compromised connective tissue function, our study has illuminated *PGR*, *NR3C1* and *HSP90AA1* –steroid pathway genes not conventionally associated with this condition, but potentially contributing to impaired endogenous progesterone effects leading to CI and/or PTB and/or reduced efficacy of progesterone administration in prolonging pregnancy in clinics (Fig 5). This discovery holds the promise of at least informed clinical decisions and improved management of CI patients; at best, it could open new avenues for therapeutic explorations. We believe that the success of identifying new CI-associated genes is rooted in meticulous patient selection. We strongly advocate for precise phenotyping of PTB patients and

Rare variant analysis identifies PGR (progesterone receptor) as a potential gene implicated in the development of cervical insufficiency

Progesterone

Progesterone receptor

Pathogenic variants in the connective tissue gene panel do not lead to cervical insufficiency in a Mendelian fashion

Genetic testing

Cervical Insufficiency

Connective tissue disfunction

Connective tissue pathways remain crucial for the development of cervical insufficiency

**Fig 5. Summary of study design and major outcomes.** We were the first to apply the Beighton/Brighton criteria to test the hypothesis linking the connective tissue dysfunction-driven nature of CI to genetics. While our analysis strengthened the association of CI with connective tissue pathways, further research is needed to explore the relationships between subclinical phenotypic expressions of connective tissue disorders and CI. Our next hypothesis is that the use of Beighton/Brighton criteria, along with the connective tissue dysfunction assessment questionnaire developed by our group, can serve as a predictive tool for CI/PTB, at least for a subset of patients. This investigation is currently underway in our group.

subgrouping based on clinical representation as a fundamental key to successful genetic and pathophysiological studies of PTB-related conditions. Collaborative efforts within the scientific community can provide more statistical power and enhance the generalizability of our observations.

## Conclusions

CI is not attributed to rare damaging variants in known genes causing connective tissue disorders in a Mendelian fashion, although this finding does not exclude the involvement of the connective tissue dysfunction pathways as a significant mechanism contributing to CI as multifactorial disorder. Rare damaging variants in the progesterone receptor (*PGR*), glucocorticoid receptor (*NR3C1*), and heat shock protein HSP 90-alpha (*HSP90AA1*), may play a role in

the pathogenesis of CI and/or PTB by interfering with progesterone's physiological effects during pregnancy.

## Supporting information

**S1 Table. Pathogenic/Likely pathogenic variants identified in controls.**
(XLSX)

**S2 Table. Genes enriched for rare damaging SNVs.**
(XLSX)

**S3 Table. Connective tissue dysfunction assessment questionnaire.**
(PDF)

**S4 Table. Connective tissue disorder panel genes.**
(CSV)

## Author Contributions

**Conceptualization:** Ludmila Voložonoka, Dmitrijs Rots, Baiba Vilne, Inga Kempa, Anna Miskova, Linda Gailīte.

**Data curation:** Ludmila Voložonoka, Līvija Bārdiņa, Zita Krūmiņa, Inga Kempa.

**Formal analysis:** Ludmila Voložonoka, Līvija Bārdiņa, Adele Rota, Baiba Vilne, Inga Kempa.

**Funding acquisition:** Ludmila Voložonoka, Inga Kempa, Anna Miskova, Linda Gailīte, Dace Rezeberga.

**Investigation:** Anna Kornete, Zita Krūmiņa, Meilė Minkauskienė, Adele Rota, Zita Strelcoviene, Anna Miskova.

**Methodology:** Ludmila Voložonoka, Līvija Bārdiņa, Anna Kornete, Zita Krūmiņa, Dmitrijs Rots, Baiba Vilne.

**Project administration:** Inga Kempa, Anna Miskova, Linda Gailīte, Dace Rezeberga.

**Resources:** Baiba Vilne, Inga Kempa, Anna Miskova, Linda Gailīte, Dace Rezeberga.

**Software:** Līvija Bārdiņa, Dmitrijs Rots, Baiba Vilne.

**Supervision:** Meilė Minkauskienė, Baiba Vilne, Inga Kempa, Anna Miskova, Linda Gailīte, Dace Rezeberga.

**Validation:** Ludmila Voložonoka.

**Visualization:** Ludmila Voložonoka.

**Writing – original draft:** Ludmila Voložonoka, Līvija Bārdiņa.

**Writing – review & editing:** Zita Krūmiņa, Dmitrijs Rots, Meilė Minkauskienė, Zita Strelcoviene, Inga Kempa, Linda Gailīte, Dace Rezeberga.

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
