## [Decision Letter · Decision Letter 0]

3 Apr 2024

PONE-D-24-05772Further Insights into the Genetics of Preterm Cervical Shortening During PregnancyPLOS ONE

Dear Dr. Volozonoka,

Thank you for submitting your manuscript to PLOS ONE. After careful consideration, we feel that it has merit but does not fully meet PLOS ONE’s publication criteria as it currently stands. Therefore, we invite you to submit a revised version of the manuscript that addresses the points raised during the review process.

We look forward to receiving your revised manuscript.

Kind regards,

Burak Bayraktar

Academic Editor

PLOS ONE

2. We note that Figure 5 and supplement 3  in your submission contain copyrighted images. All PLOS content is published under the Creative Commons Attribution License (CC BY 4.0), which means that the manuscript, images, and Supporting Information files will be freely available online, and any third party is permitted to access, download, copy, distribute, and use these materials in any way, even commercially, with proper attribution. For more information, see our copyright guidelines: http://journals.plos.org/plosone/s/licenses-and-copyright.

a. You may seek permission from the original copyright holder of Figure 5 and supplement 3 to publish the content specifically under the CC BY 4.0 license. 

Additional Editor Comments:

Editor:

“We conducted whole exome sequencing (WES) analysis on 114 patients diagnosed with a short cervix, CI or a history of CI in previous pregnancies.” How did you choose the cases? More precisely, was this study a prospective study on short cervix? Or were patients who underwent WES examination for other indications retrospectively investigated for short cervix?

Can you include patient’s demographics? Smoking, uterine anomaly, systemic diseases, etc. was the excluded?

I guess it means you found no significant genome changes?

Reviewers' comments:

Reviewer's Responses to Questions

**Comments to the Author**

1. Is the manuscript technically sound, and do the data support the conclusions?

Reviewer #1: No

2. Has the statistical analysis been performed appropriately and rigorously? 

Reviewer #1: No

3. Have the authors made all data underlying the findings in their manuscript fully available?

Reviewer #1: No

4. Is the manuscript presented in an intelligible fashion and written in standard English?

Reviewer #1: Yes

5. Review Comments to the Author

Reviewer #1: Study of important and relevant themes for the magazine. The main objective of the study was to deepen discussions about the genetic factors that contribute to IC.

The study brings relevant results, but its biggest problem is that it does not present a methods session. There is a very brief paragraph in the discussion commenting on what was carried out in the study. I strongly suggest that the authors insert a methods session into the study involving study design, setting (where the study was carried out), population studied, inclusion and exclusion criteria, how these participants were recruited, procedures carried out for data collection, analysis of data, ethics committee, etc. Without this session, it is not possible to evaluate the study and say whether it was carried out correctly or whether there were biases.

Another point that I think is important to mention is that the title is not eye-catching and does not arouse the reader's curiosity.

I leave as a suggestion: Unraveling the Genetic Landscape of Cervical Insufficiency: Insights into Connective Tissue Dysfunction and Hormonal Pathways

In the summary, I suggest clearly and objectively stating the objective of the study and also structuring a methods session.

Keywords: lots of keywords. Choose a maximum of five.

In the introduction, it is important to provide a little more detail about the reasons why the PTB is a public health concern. What are its implications?

Furthermore, what were the motivations for conducting the study? What are the hypotheses? What new does the study bring? What knowledge gaps do you aim to fill? The justification for the study needs to be clearer and better defined in the introduction

Regarding the results and discussion, I have no comments at first, I would need to know more details about how the study was conducted (methods) to be able to assess whether the results were reported and discussed appropriately.

6. PLOS authors have the option to publish the peer review history of their article (what does this mean?). If published, this will include your full peer review and any attached files.

Reviewer #1: No

---

## [Author Response · Author response to Decision Letter 0]

17 May 2024

Dear Burak Bayraktar,

Dear Reviewer,

Thank you very much for processing and reviewing our manuscript. We have addressed all the comments and provided point-by-point answers below, incorporating the suggested changes into the manuscript.

The provided line numbers are from the unmarked version of revised paper without tracked changes.

Editor:

Question: “We conducted whole exome sequencing (WES) analysis on 114 patients diagnosed with a short cervix, CI or a history of CI in previous pregnancies.” How did you choose the cases? More precisely, was this study a prospective study on short cervix? Or were patients who underwent WES examination for other indications retrospectively investigated for short cervix?

Answer: The process of enrolling patients in the study is described in the Methods section (lines 483-502). Briefly, patients undergoing routine antenatal care were examined for a short cervix, and those requiring treatment for cervical shortening were recruited. This study was conducted as a prospective longitudinal cohort study, as detailed in the Methods section (lines 484).

Question: Can you include patient’s demographics? Smoking, uterine anomaly, systemic diseases, etc. was the excluded?

Answer: Some patient demographics, including ethnicity, age, and parity, are described in the Results section (lines 128-132). Additional phenotype data was collected using the connective tissue dysfunction assessment questionnaire, as detailed in the Methods section (lines 497-499). We have now supplemented the Methods section with the exclusion criteria, which also includes uterine anomalies. Notably, systemic diseases (unless part of a genetic syndrome) and smoking were not exclusion criteria.

Question: I guess it means you found no significant genome changes?

Answer: None of the patients in our study were found to have monogenic connective tissue disorder. Instead, our main finding was the identification of rare damaging variant enrichment in steroid hormone receptor pathway genes, with variants in PGR, HSP90AA1, and NR3C1 reaching genome-wide statistical significance even after correction for multiple testing (as indicated in lines 262-263 of the Results section). 

Reviewer #1: 

Question: Study of important and relevant themes for the magazine. The main objective of the study was to deepen discussions about the genetic factors that contribute to IC.

The study brings relevant results, but its biggest problem is that it does not present a methods session. There is a very brief paragraph in the discussion commenting on what was carried out in the study. I strongly suggest that the authors insert a methods session into the study involving study design, setting (where the study was carried out), population studied, inclusion and exclusion criteria, how these participants were recruited, procedures carried out for data collection, analysis of data, ethics committee, etc. Without this session, it is not possible to evaluate the study and say whether it was carried out correctly or whether there were biases.

Answer: Thank you sincerely for dedicating your time and effort to reviewing our manuscript. We apologize for any confusion regarding the Methods section's visibility. You can find the Methods section starting with line 474. The Methods placement after the Discussion is in line with the formatting guidelines stipulated by PLOS One.

Question: Another point that I think is important to mention is that the title is not eye-catching and does not arouse the reader's curiosity. I leave as a suggestion: Unraveling the Genetic Landscape of Cervical Insufficiency: Insights into Connective Tissue Dysfunction and Hormonal Pathways

Answer: Thank you very much for this nice suggestion, with pleasure we are accepting this new title.

Question: In the summary, I suggest clearly and objectively stating the objective of the study and also structuring a methods session.

Answer: The primary aim of our study is stated in lines 115-116, and we have revised it for clarity.

Question: Keywords: lots of keywords. Choose a maximum of five.

Answer: We understand the concern regarding the abundance of keywords. However, PLOS One encourages the inclusion of multiple keywords to enhance the discoverability of published articles. Therefore, we prefer to retain the current keywords.

Question: In the introduction, it is important to provide a little more detail about the reasons why the PTB is a public health concern. What are its implications?

Answer: We have briefly addressed the paragraph on PTB (please see lines 46-50). However, we chose not to significantly expand this paragraph as considerable literature already exists discussing the social and economic concerns of PTB, as referenced in this paragraph. Given that the primary focus of our article is the genetics of cervical insufficiency, we opted not to shift the primary focus of the study.

Question: Furthermore, what were the motivations for conducting the study?

Answer: Our primary motivation for studying the genetics of CI stems from clinical needs, as pregnancy outcomes in this patient group are particularly concerning (see lines 97-99 and 276-278 in the Discussion section). The genetics of PTB and particularly CI remain poorly understood, with existing literature often lacking clinical subgrouping. We aimed to address this gap by focusing specifically on the CI patient group, anticipating insights into specific genes and molecular pathways, particularly related to the connective tissue hypothesis (112-114).

Question: What are the hypotheses?

Answer: As you may understand, the hypothesis for this study stems from our previous pilot study where we identified variants in extracellular matrix-related genes in patients with cervical insufficiency. This led us to hypothesize that CI might represent a subtle form of connective tissue disorder (please refer to lines 108-114).

Question: What new does the study bring?

Answer: While we understand that some studies choose to highlight findings in the Introduction, we have opted to reserve the anticipation of study results for the Results and Discussion sections.

Question: What knowledge gaps do you aim to fill?

Our study aims to address significant knowledge gaps in the field of cervical insufficiency. Currently, there are no aetiological treatments available for this condition, largely due to limited understanding of its molecular mechanisms and the neglect of CI genetics in the scientific community. Our goal is to decipher the molecular pathways underlying CI to contribute to a better understanding of its pathogenesis.

Question: The justification for the study needs to be clearer and better defined in the introduction.

Answer: Thank you very much for the detailed analysis of our Introduction section and providing valuable insights. We have made substantial changes in the Introduction to better clarify the justification for conducting our study.

As outlined in the Introduction, there is limited understanding of the genetics of cervical insufficiency, and our pilot study was among the first to apply next-generation sequencing to this patient group, for the current study we collected the largest cohort of patients for whole exome sequencing. We aimed to validate the long-standing link between connective tissue dysfunction and CI through genetic studies and developed the connective tissue dysfunction assessment as a comprehensive clinical tool for this purpose.

Unexpectedly, our study also revealed significant findings related to the progesterone pathway, despite the modest sample size. We hope you will find this aspect of the study particularly intriguing.

Question: Regarding the results and discussion, I have no comments at first, I would need to know more details about how the study was conducted (methods) to be able to assess whether the results were reported and discussed appropriately.

Answer: Thank you for your initial assessment. We understand that a thorough evaluation of the Results and Discussion sections depends on a comprehensive understanding of the study methods. We hope that our Methods section will provide the necessary details, and we look forward to your feedback on the manuscript once it is complete.

Once again, we hope our clarifications have addressed all current concerns, and we remain open to any further suggestions or feedback you may have.

Thank you once again for your valuable input.

Kind regards,

Authors

---

## [Decision Letter · Decision Letter 1]

15 Jul 2024

PONE-D-24-05772R1Unravelling the Genetic Landscape of Cervical Insufficiency: Insights into Connective Tissue Dysfunction and Hormonal PathwaysPLOS ONE

Dear Dr. Volozonoka,

Thank you for submitting your manuscript to PLOS ONE. After careful consideration, we feel that it has merit but does not fully meet PLOS ONE’s publication criteria as it currently stands. Therefore, we invite you to submit a revised version of the manuscript that addresses the points raised during the review process.

We look forward to receiving your revised manuscript.

Kind regards,

Burak Bayraktar

Academic Editor

PLOS ONE

Reviewers' comments:

Reviewer's Responses to Questions

**Comments to the Author**

1. If the authors have adequately addressed your comments raised in a previous round of review and you feel that this manuscript is now acceptable for publication, you may indicate that here to bypass the “Comments to the Author” section, enter your conflict of interest statement in the “Confidential to Editor” section, and submit your "Accept" recommendation.

Reviewer #2: All comments have been addressed

Reviewer #3: All comments have been addressed

2. Is the manuscript technically sound, and do the data support the conclusions?

Reviewer #2: Partly

Reviewer #3: Yes

3. Has the statistical analysis been performed appropriately and rigorously? 

Reviewer #2: No

Reviewer #3: Yes

4. Have the authors made all data underlying the findings in their manuscript fully available?

Reviewer #2: Yes

Reviewer #3: Yes

5. Is the manuscript presented in an intelligible fashion and written in standard English?

Reviewer #2: Yes

Reviewer #3: Yes

6. Review Comments to the Author

**Reviewer #2: **Dear Authors,

this is an interesting prospective longitudinal cohort study, trying to identify the molecular causes of cervical insufficiency. This one represents one of the origins of the main cause itself of fetal morbidity and mortality in Obstetrics: Preterm Birth. I read with great interest the Manuscript, which falls within the aim of this Journal. It is the revision number two for this journal. Several of the corrections required have been applied. In my honest opinion, some major improvements and corrections are still needed.

Title- I see the corrections applied

Abstract:

The abstract should better be structured for sake of clarity. It should contain background, Objective, methods, Results and Conclusion. More insights regarding the methods are already needed

Introduction:

Improvements have been made according to the previous reviewer advices, but a descriptive paragraph of what preterm birth is, should be written. I have no objections for the rest of the introduction

Results: This paragraph appears too repetitive and it should not just be the list of the collected data. Numbers should be presented in a systematic and goal-oriented manner.

Discussion:

The discussion should elucidate and find possible explanations for the research. It should be revised in the light of the methods.

Conclusions:

Rename this section.

It should be more consistent and summarize your main findings, clinical implications and suggest areas for further research.

Avoid overgeneralization.

Materials and Methods:

There are some improvement compared to the first version, but is still not enough to make the manuscript scientifically sound. There is no chance to promote this kind of study lacking an appropriate Materials and Methods section.

**Reviewer #3:** Dear authors,

Thank you for revising the manuscript considering the previously suggestions from the reviewers in the first review round.

All the comments of the previously reviewers have been addressed by the authors and the manuscript was improved.

I strongly suggest to the authors to move the Methods, according to the Plos One submission guidelines called "Materials and Methods", after the Introduction section.

The "Materials and Methods" in the Plos One is located in the Middle section, after the introduction.

Please verify the manuscript organization located in this page: https://journals.plos.org/plosone/s/submission-guidelines

I do not have additional comments on the revision in this manuscript version.

7. PLOS authors have the option to publish the peer review history of their article (what does this mean?). If published, this will include your full peer review and any attached files.

Reviewer #2: No

Reviewer #3: No

---

## [Author Response · Author response to Decision Letter 1]

21 Aug 2024

Dear Burak Bayraktar,

Dear Reviewers,

Thank you very much for processing and reviewing our manuscript. We have addressed all the comments and provided point-by-point answers below, incorporating the suggested changes into the manuscript.

The provided line numbers are from the unmarked version of revised paper without tracked changes.

Reviewer #2: 

Dear Authors,

this is an interesting prospective longitudinal cohort study, trying to identify the molecular causes of cervical insufficiency. This one represents one of the origins of the main cause itself of fetal morbidity and mortality in Obstetrics: Preterm Birth. I read with great interest the Manuscript, which falls within the aim of this Journal. It is the revision number two for this journal. Several of the corrections required have been applied. In my honest opinion, some major improvements and corrections are still needed.

Title- I see the corrections applied

Abstract:

The abstract should better be structured for sake of clarity. It should contain background, Objective, methods, Results and Conclusion. More insights regarding the methods are already needed

Answer: Dear Reviewer, thank you very much for your time and valuable feedback on improving our manuscript. We have restructured the Abstract to include Background, Methods, Results and Conclusion. We have enhanced the text to provide more details and clarity.

Introduction:

Improvements have been made according to the previous reviewer advices, but a descriptive paragraph of what preterm birth is, should be written. I have no objections for the rest of the introduction

Answer: Please refer to lines 48-79 of the Introduction, which includes a sub-section titled ‘Preterm birth: one outcome – multiple aetiologies and distinct molecular pathways’ (line 53) providing prevalence, etiological pathways, contributing factors, and current treatment options of PTB.

Results: This paragraph appears too repetitive and it should not just be the list of the collected data. Numbers should be presented in a systematic and goal-oriented manner.

Answer: We have revised the Results section to minimize repetition and present data systematically. The structure focuses on:

1. Variant analysis in genes associated with connective tissue disorders, based on ACGS guidelines, to explore the hypothesis that CI might reflect a subtle connective tissue disorder. We also correlated genetic findings with connective tissue dysfunction assessment results (lines 317-360).

2. Genome-wide (a hypothesis-free) rare variant burden test that identified PGR, HSP90AA1, and NR3C1 genes as potentially significant in CI development (lines 361-417).

3. Pathway enrichment analyses performed both in a targeted manner and genome-wide (lines 418-433).

We believe our revisions address the concern about repetitiveness and present the data in a clear manner.

Discussion:

The discussion should elucidate and find possible explanations for the research. It should be revised in the light of the methods.

Answer: The Discussion section provides detailed explanations and contextualize our findings based on the methodologies used. The discussion is organized into sub-sections corresponding to the different perspectives of our analysis (i.e., connective tissue perspective, hormonal pathway perspective). Each sub-section addresses our findings in the context of existing literature and aims to elucidate possible explanations.

If Reviewer have specific concerns or inquiries regarding any part of the discussion, we would be happy to address them.

Conclusions:

Rename this section.

It should be more consistent and summarize your main findings, clinical implications and suggest areas for further research.

Avoid overgeneralization.

Answer: We have renamed and revised the Conclusions section to ensure it provides a clear and concise summary of our main findings, avoiding overgeneralization.

Materials and Methods:

There are some improvement compared to the first version, but is still not enough to make the manuscript scientifically sound. There is no chance to promote this kind of study lacking an appropriate Materials and Methods section.

Answer: The Methods section has now been moved after the Introduction and provides a detailed account of all steps performed during the study. If the Reviewer has specific concerns or requires further clarification on any aspect of this section, we would be happy to address them. We have aimed to ensure that all procedures involving patient and genomic data are thoroughly described.

Reviewer #3: Dear authors,

Thank you for revising the manuscript considering the previously suggestions from the reviewers in the first review round.

All the comments of the previously reviewers have been addressed by the authors and the manuscript was improved.

I strongly suggest to the authors to move the Methods, according to the Plos One submission guidelines called "Materials and Methods", after the Introduction section.

The "Materials and Methods" in the Plos One is located in the Middle section, after the introduction.

Please verify the manuscript organization located in this page: https://journals.plos.org/plosone/s/submission-guidelines

I do not have additional comments on the revision in this manuscript version.

Answer: Thank you very much for your time and efforts in reviewing our manuscript. We have moved the Methods section immediately after the Introduction.

Once again, we hope our clarifications have addressed all current concerns, and we remain open to any further suggestions or feedback you may have.

Thank you once again for your valuable input.

Kind regards,

Authors

---

## [Editor Report · Decision Letter 2]

6 Sep 2024

Unravelling the Genetic Landscape of Cervical Insufficiency: Insights into Connective Tissue Dysfunction and Hormonal Pathways

PONE-D-24-05772R2

Dear Dr. Volozonoka,

We’re pleased to inform you that your manuscript has been judged scientifically suitable for publication and will be formally accepted for publication once it meets all outstanding technical requirements.

Kind regards,

Burak Bayraktar

Academic Editor

PLOS ONE

---

## [Editor Report · Acceptance letter]

10 Sep 2024

PONE-D-24-05772R2 

PLOS ONE

Dear Dr. Volozonoka, 

I'm pleased to inform you that your manuscript has been deemed suitable for publication in PLOS ONE. Congratulations! Your manuscript is now being handed over to our production team.

Kind regards, 

on behalf of

Dr. Burak Bayraktar 

Academic Editor

PLOS ONE